# Compression and Recovery of 3D Broad-Leaved Tree Point Clouds Based on Compressed Sensing

**Renjie Xu [1], Ting Yun [1] , Lin Cao [2] and Yunfei Liu [1],***

[1]  College of Information Science and Technology, Nanjing Forestry University, Nanjing 210037, China;
    xurenjie@njfu.edu.cn (R.X.); njyunting@gmail.com (T.Y.)
[2]  Co-Innovation Center for Sustainable Forestry in Southern China, Nanjing Forestry University,
    Nanjing 210037, China; lincao@njfu.edu.cn
*   Correspondence: lyf@njfu.com.cn; Tel.: +86-139-1389-5117

**Abstract:** The terrestrial laser scanner (TLS) has been widely used in forest inventories. However, with increasing precision of TLS, storing and transmitting tree point clouds become more challenging. In this paper, a novel compressed sensing (CS) scheme for broad-leaved tree point clouds is proposed by analyzing and comparing different sparse bases, observation matrices, and reconstruction algorithms. Our scheme starts by eliminating outliers and simplifying point clouds with statistical filtering and voxel filtering. The scheme then applies Haar sparse basis to thin the coordinate data based on the characteristics of the broad-leaved tree point clouds. An observation procedure down-samples the point clouds with the partial Fourier matrix. The regularized orthogonal matching pursuit algorithm (ROMP) finally reconstructs the original point clouds. The experimental results illustrate that the proposed scheme can preserve morphological attributes of the broad-leaved tree within a range of relative error: 0.0010%–3.3937%, and robustly extend to plot-level within a range of mean square error (MSE): 0.0063–0.2245.

**Keywords:** terrestrial laser scanner; forest inventories; point cloud; broad-leaved tree; compressed sensing; morphological attributes; plot-level

## 1. Introduction

A terrestrial laser scanner (TLS) can provide three-dimensional (3D) co-ordinates of sampled points at a pre-defined sampling interval. They have high measurement accuracy and data acquisition efficiency, and are suitable for capturing large scenes with relatively low expenditure. TLS has been used to obtain 3D observations on tree surfaces to study morphological attributes [1–5]. However, the sampled data from TLS is very dense and with considerable redundancy. The vast volume of point clouds poses great challenges in real-time processing, storage, display, and transmission. Therefore, it is necessary to compress massive point clouds while maintaining a certain accuracy.

For a practical compression algorithm of point cloud, the following criterions must be satisfied: (1) high compression rate, i.e., the number of point clouds should be minimized subject to a distortion bound; (2) the simplified point clouds can meet the accuracy requirements of target applications; (3) the algorithm needs to be computationally efficient.

In recent years there has been considerable research on point cloud compression. Digne et al. [6] exploited the self-similarity of the underlying shapes to create a particular dictionary on which the local neighborhoods will be sparsely represented; thus, allowing for a lightweight representation of the total surface. Wang et al. [7] applied planar reflective symmetry analysis to identify a primary symmetry plane and three orthogonal projection planes given a point-based model. By analyzing the characteristics of the projection on these planes, different and specific coders were employed.

Zhang et al. [8] employed mean-shift clustering to gather similar spatial points into many homogeneous blocks, which were fitted into surfaces by Random Sample Consensus (RANSAC) algorithm. The color (RGB) information of each point was then replaced by the average RGB values in the grid. Finally, DCT (Discrete Cosine Transform) was performed on grids for sparse representation. Ahn et al. [9] proposed a hybrid range image coding algorithm, which predicted the radial distance of each pixel adaptively by utilizing the previously encoded neighborhood in the range image domain, height image domain, and 3D domain. Queiroz et al. [10] combined hierarchical transform and arithmetic coding to compress the color information of point clouds, which ensured the whole transform was adaptive, non-expansive and orthogonal. Thanou et al. [11] exploited temporal correlation between consecutive point clouds to develop a compression scheme for 3D point cloud sequences. Navarrete et al. [12] took a non-supervised learning algorithm based on Gaussian Mixture Models (GMMs) together with the Expectation-Maximization (EM) algorithm to compress a 3D point cloud, which excelled at reducing the size of data and enhancing efficiency. Rente et al. [13] proposed an efficient lossy coding mechanism for the geometry of static point clouds. It applied an octree-based approach for a base layer and a graph-based transform approach for the enhancement layer where an inter-layer residual was coded. Imdad et al. [14] compressed 3D point clouds by representing implicit surfaces with polynomial equations of degree one, which retained geometric information of scene with low storage complexity. Cui et al. [15] proposed a palette-based compression method for color information of 3D point clouds. It created a color palette according to the spatial redundancy among color attribute data, and applied K-means clustering method to remove redundancy among adjacent color data. However, existing compression algorithms [16–32] of point clouds have several weaknesses: (1) low computational efficiency; (2) high time cost; (3) inability to handle complex point clouds, and (4) the need for full sampling.

It is well known that due to the complex morphological structures and massiveness of point clouds captured from trees or forests, the traditional methods that perform compression after full sampling are very inefficient and time-consuming. Moreover, most compression algorithms cannot deal with point clouds with high complexity, such as trees or forests. The emergence of compressed sensing [33–35] has brought a new breakthrough to the Shannon–Nyquist sampling theorem. It acquires data by random sampling with very few sampling points to achieve the same effect as full sampling and has been widely used in Magnetic Resonance Imaging (CS-MRI) [36], high-speed video camera [37], compressive spectral imaging system [38], single-pixel camera [39], etc.

In this paper, a novel scheme based on compressed (PDF S1) sensing for acquiring broad-leaved tree point clouds is proposed. Firstly, voxel filtering and statistical filtering are employed to simplify point clouds and remove outliers, respectively. Secondly, the spatial coordinate data (XYZ) are divided into three one-dimensional data to be processed in parallel since compressed sensing is not able to process three-dimensional data directly. In addition, since the one-dimensional data is too large to be processed directly, it is arranged into matrix and processed by columns in parallel. A suitable sparse basis is selected for sparse representation of data according to the characteristics of broad-leaved tree point clouds, and partial Fourier matrix is applied to down-sample the sparse data due to its RIP (Restricted Isometry Property) [33,40]. Finally, the most suitable ROMP (Regularized Orthogonal Matching Pursuit) algorithm [41] is selected to reconstruct the data accurately.

## 2. Materials and Methods

### 2.1. Study Area and Equipment Introduction

Point clouds (a)–(g) were collected at the Laoshan National Forest Park (32°5′36″–32°7′4″ N, 118°36′2″–118°36′55″ E) in Nanjing City, Jiangsu Province (31°14′–32°37′ N, 118°22′–119°14′ E), and point cloud (h) was collected at the rubber tree plantation in Danzhou City (19°11′–19°52′ N, 108°56′–109°46′ E), Hainan Province (18°10′–20°10′ N, 108°37′–111°03′ E), using the popular high

definition Velodyne HDL-32E LiDAR sensor (Velodyne Lidar, Inc, US). The configuration is given in Table 1.

**Table 1.** Configuration of Velodyne HDL-32E LiDAR sensor.

| Technical Parameters | Technical Specifications |
| --- | --- |
| Measurement Distance | 80 m (min) to 100 m (max) |
| Points per second | Up to 695,000 |
| Scanning accuracy | <2 cm |
| Field of view | Horizontal: 0° to 360° |
| | Vertical: −30.67° to +10.67° |

### 2.2. Point Clouds

The collected data, including point clouds from a single tree (e.g., *cherry tree*, *papaya tree*, *poplar*), as well as from plots (e.g., *sapindus plot*, *poplar plot*, *rubber tree plot*) are illustrated in Figure 1. The flow chart of the proposed scheme is given in Figure 2. The procedure consists of data preprocessing, sparse transformation, data down-sampling, and data reconstruction.

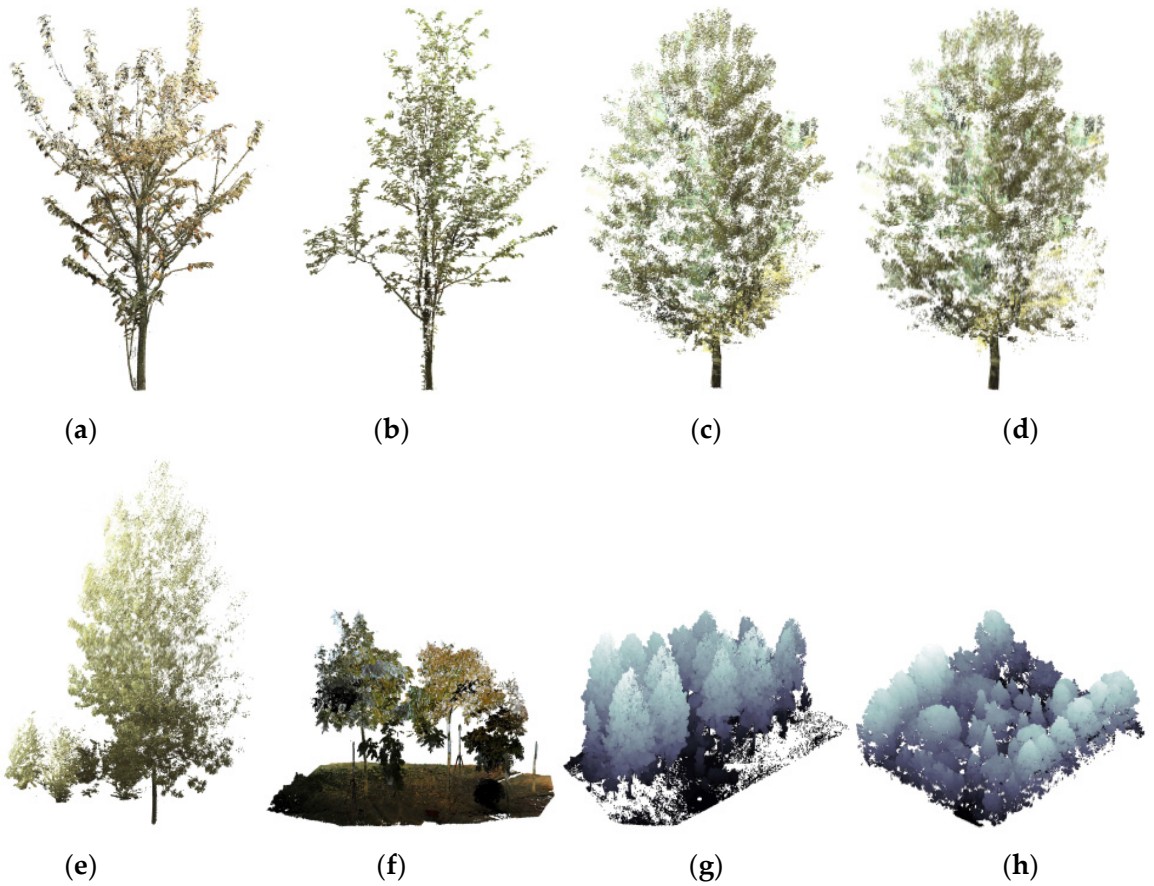

(**a**)　　　　　(**b**)　　　　　(**c**)　　　　　(**d**)

(**e**)　　　　　(**f**)　　　　　(**g**)　　　　　(**h**)

**Figure 1.** The collected point clouds from Velodyne HDL-32E LiDAR sensor. Single-tree point clouds: (**a**) *Cherry tree*, (**b**) *Papaya tree*, (**c**) *Poplar 1*, (**d**) *Poplar 2*, (**e**) *Poplar 3* with shrubs, and plot point clouds: (**f**) *Sapindus plot*, (**g**) *Poplar plot*, (**h**) *Rubber tree plot*.

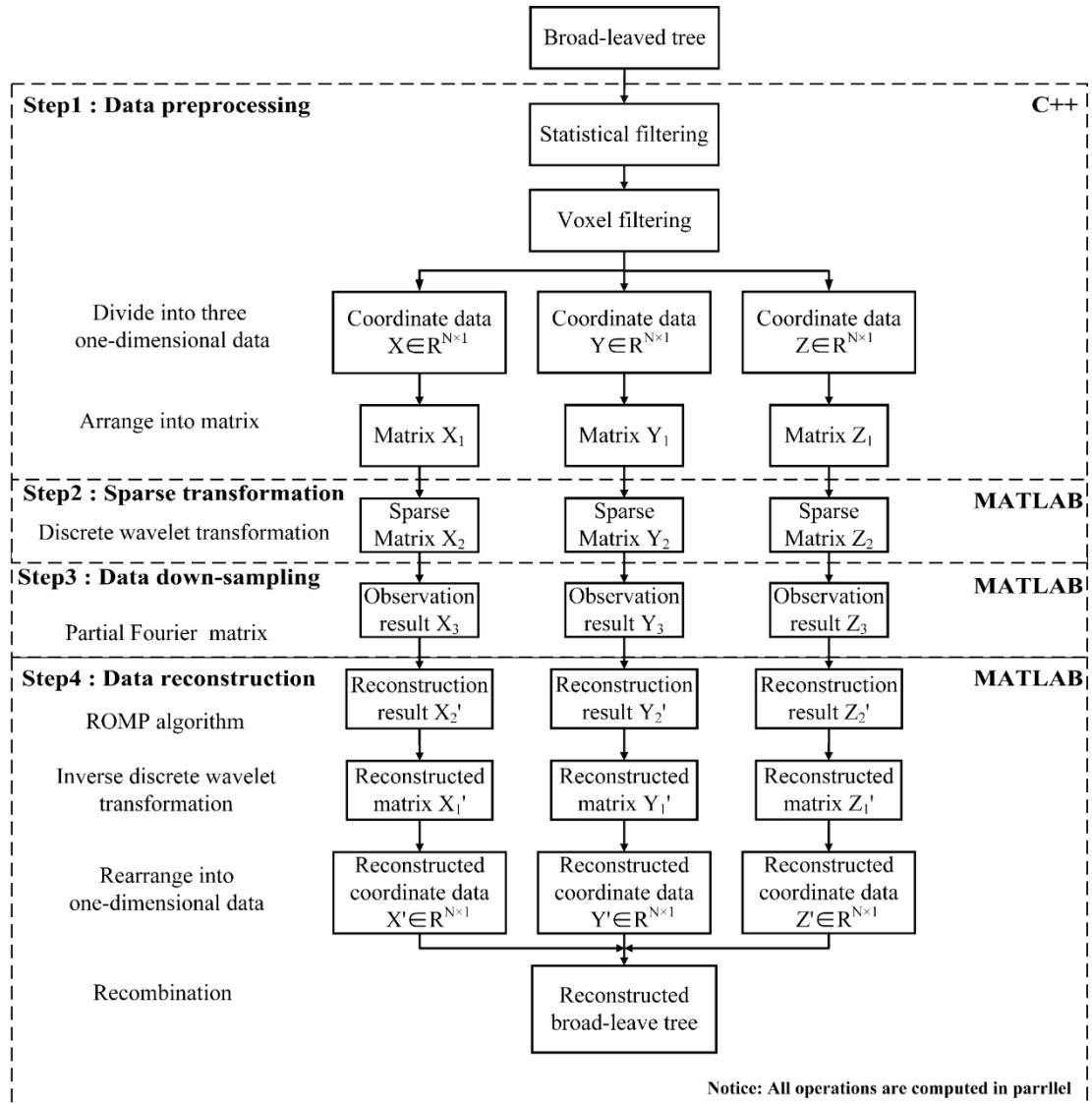

**Figure 2.** A flow chart of the proposed scheme. Note that ROMP denotes the Regularized Orthogonal Matching Pursuit.

*2.3. Date Preprocessing*

In the first step, voxel filtering and statistical filtering are applied to simplify the tree point clouds acquired from TLS and remove outliers. Due to the massive volume of tree point clouds, the observation matrix and the sparse transform matrix are very large and difficult to process. If the data is not arranged appropriately, direct computation will consume hundreds of gigabytes of memory. Furthermore, compressed sensing is based on one-dimensional data processing, and thus cannot be applied to 3D point clouds directly. Therefore, we first split the spatial coordinate data (XYZ) into three separate one-dimensional data and then arrange them into three matrices. Take X-coordinate as an example (Figure 3). Every $2^N$ points compose a column of the resulting matrix, namely, $1 \sim 2^N$ points compose the first column, $2^N + 1 \sim 2^{N+1}$ points compose the second column and so on. Note that if the number of points in the last column is less than $2^N$, the remaining elements are set to be zero. The procedure of preprocessing can greatly save time and memories. In this paper, N is set to 8 due to its superior performance.

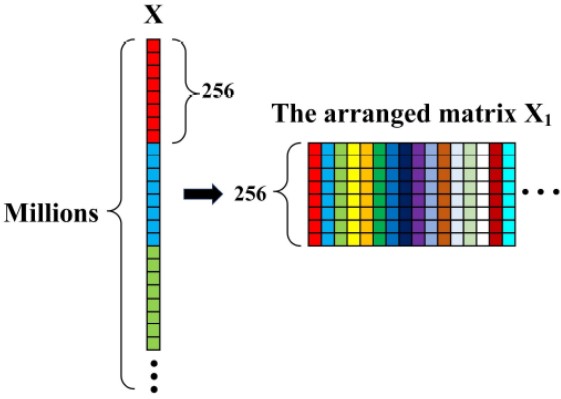

**Figure 3.** Date arrangement where *N* = 8.

### 2.4. Sparse Transformation

After data preprocessing, three spatial coordinate matrices $X_1$, $Y_1$, $Z_1$ are obtained. By observing each column of $X_1$, $Y_1$, $Z_1$ matrix, we find that most data have similar morphological characteristics: step-wise pattern (Figure 4a–c), which motivates us to apply DWT (Discrete Wavelet Transform) based on Haar wavelet (Figure 4d). The sparse matrix $X_2$ can be obtained by:

$$X_2 = \Psi X_1 \tag{1}$$

where $\Psi$ denotes the DWT matrix (Figure 5). Experimental results in Section 3.1 confirms that Haar wavelet is the most suitable choice among common wavelets like Daubechies (db), Symlets (sym), Coiflets (coif), Biorthogonal (Bior), Reverse Biorthogonal (rbio) and Fejer-Korovkin (fk).

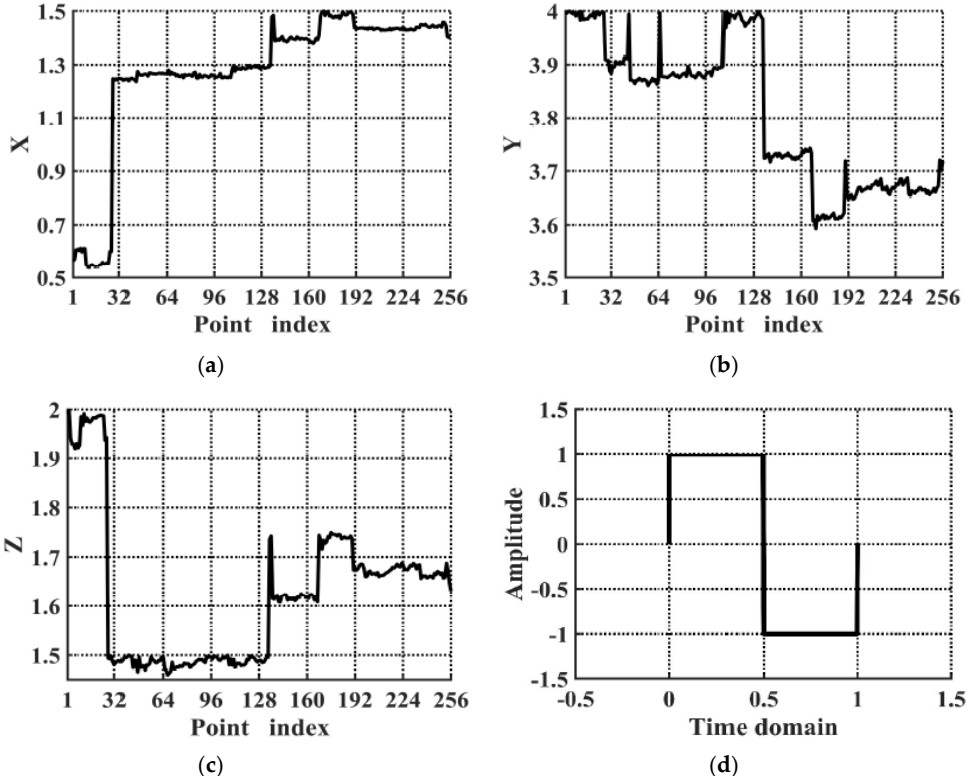

**Figure 4.** The distribution of the *Cherry tree* point cloud: (**a**) An example distribution of matrix $X_1$; (**b**) An example distribution of matrix $Y_1$; (**c**) An example distribution of matrix $Z_1$. (**d**) The waveform of Harr wavelet.

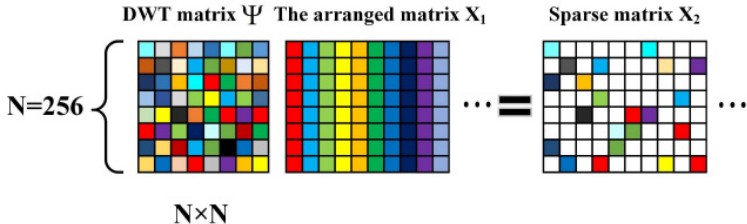

**Figure 5.** Sparse Transformation.

## 2.5. Data Down-Sampling

According to the compressed sensing theory based on sparse structure, the original data can be accurately recovered from only a few sampling points provided that the data is sparse. Moreover, in order to obtain the unique reconstructed solution, the observation matrix must satisfy the RIP condition. In this paper, a partial Fourier matrix is applied to down-sample the sparse data. We multiply partial Fourier matrix to each column of the sparse matrix $X_2$ (Figure 6) to obtain the observation result $X_3$:

$$X_3 = \Phi X_2, \tag{2}$$

where $\Phi$ denotes the observation matrix.

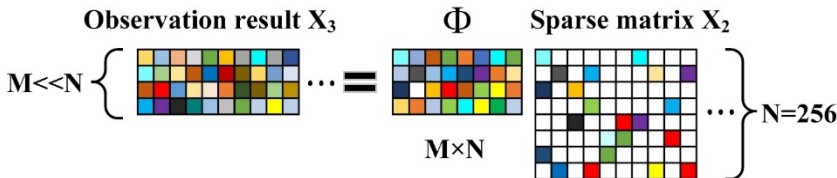

**Figure 6.** The process of observation.

## 2.6. Data Reconstruction

Since $M \ll N$, the scale of observation result $X_3$ is much smaller than that of $X_2$, which is convenient for storage and transmission. To recover the sparse matrix $X_2'$ from $X_3$, we apply the ROMP (Algorithm 1) provided that the observation result $X_3$ and observation matrix $\Phi$ are known. By inverse DWT, the arranged matrix $X_1'$ is computed. Finally, the arranged matrix $X_1'$ is rearranged into one-dimensional data $X'$, which means the data reconstruction is completed (Figure 7). The inverse DWT is given by:

$$X_1' = \Psi^{-1} X_2' \tag{3}$$

---

**Algorithm 1.** Regularized Orthogonal Matching Pursuit Algorithm (ROMP).

---

**INPUT:**     Sensing matrix $\Theta = \Phi\Psi$, observation result $y \in \mathbb{R}^N$ and sparse level K
**OUTPUT:**   Reconstruction result $\hat{x}$

---

1: Index set $\Lambda = \varnothing$;
2: Residual $r_0 = Y$;
3: The number of iterations $t = 1$;
4: For $t \le K$ and $\| \Lambda \|_0 \le 2K$ do
5: Compute the correlation coefficient $u = \left\{ u_j \, \middle| \, u_j = |< r, \varphi_j >|, j = 1, 2, \ldots N \right\}$ and choose a set J of the K biggest nonzero coordinates in the magnitude of u, or all of its nonzero coordinates, whichever set is smaller;
6: Among all subsets $J_0 \subset J$ with comparable coordinates $|u(i)| \le 2|u(j)|$ $i, j \in J_0$, choose $J_0$ with maximal energy $\| u|_{J_0} \|_2$;
7: Update $\Lambda = \Lambda \bigcup J_0$ and the support set $\Phi_\Lambda$;
8: Compute $\hat{x} = \underset{x \in R^\Lambda}{\text{argmin}} \| Y - \Phi_\Lambda x \|_2$ and update $r_{new} = Y - \Phi_\Lambda \hat{x}$, $t = t + 1$;
9: End for

---

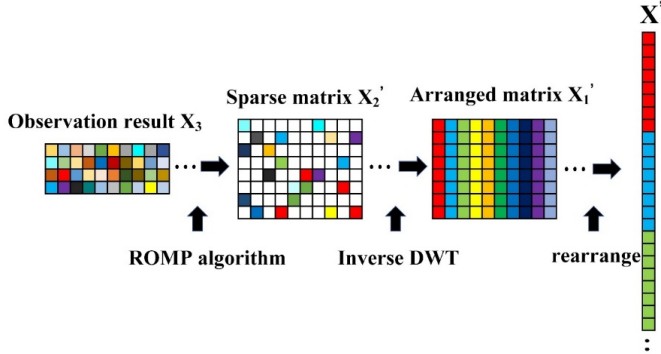

**Figure 7.** The process of data reconstruction.

## 3. Results

In this section, we first compare several classical wavelet bases, observation matrices, and reconstruction algorithms with the proposed scheme (Excel S1). Then, we report the experimental results of the proposed scheme on real-world point clouds. Our algorithm has been implemented in MATLAB and C++. The experimental platform is Inter® Core™ I5-9400F CPU @ 2.90GHZ, 16GB RAM, Windows 10 operating system.

We utilize the mean square errors (MSE) of reconstructed data as an evaluation metric:

$$\text{MSE} = \frac{\|X - X'\|_2^2}{N} \tag{4}$$

where X denotes the original data, X' denotes the reconstructed data, and N is the length of the data. Let M denote the number of compressive measurements. The compression ratio can be represented as $M/N$, also called the sampling rate.

### 3.1. Comparison

For comparison, we choose several classical wavelet bases, observation matrices, and reconstruction algorithms based on their suitability for tree point clouds compression. Note that the parameters of each experiment have been tuned to the optimal values. Each data point in the figures is the numerical mean of multiple (ten times) experiments.

Firstly, we compare the reconstruction accuracy using different wavelet bases. In order to ensure the validity of the experiment, we only change the wavelet and keep the observation matrix and reconstruction algorithm the same.

As can be seen in Figure 8, the scheme based on the Haar wavelet consistently achieves the lowest reconstruction error with comparable reconstruction time as the other wavelet bases over multiple point clouds. It can thus be concluded that the Haar wavelet is a suitable sparse basis for broad-leaved tree point clouds.

Secondly, we compare the reconstruction accuracy using different observation matrices. In the experiment, only the sparse transform matrix is changed, while the sparse basis and reconstruction algorithm remain the same.

Figure 9 shows that the reconstruction errors of partial Fourier matrix are much smaller than those of other observation matrices. The improved accuracy, however, comes at the cost of doubling the reconstruction time. For broad-leaved trees, the reconstruction error has a significant impact on the preservation of tree morphological attributes. Therefore, as long as the computation time is acceptable, partial Fourier matrix is a better choice due to its high accuracy.

Finally, we compare the effect of reconstruction algorithms with only changing the algorithms during the experiment.

Lastly, we compare the ROMP algorithm with other reconstruction algorithms. From Figure 10, it can be observed that ROMP algorithm achieves lower reconstruction errors and it takes much less

time than other algorithms. It only takes a few seconds on a 100,000-point point cloud and almost 15 seconds on a 1,000,000-point point cloud. Thus, it is reasonable to conclude that ROMP algorithm is superior in recovering broad-leaved tree point clouds.

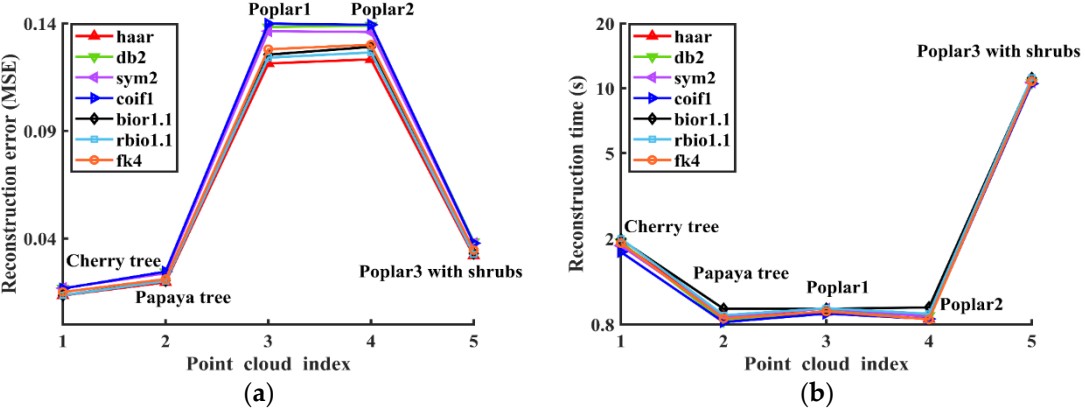

**Figure 8.** Comparison of wavelet bases at compression ratio: 40%. (**a**) Reconstruction error (mean square error (MSE)). (**b**) Reconstruction time.

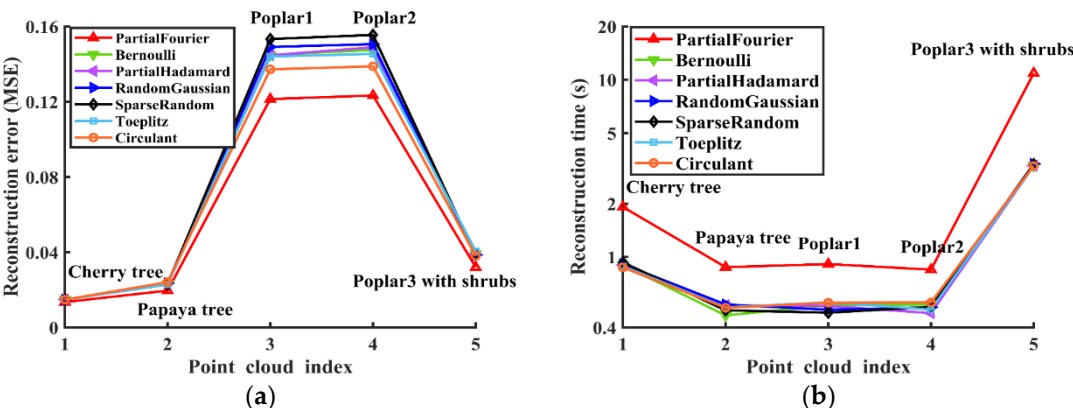

**Figure 9.** Comparison of observation matrices at compression ratio: 40%. (**a**) Reconstruction error (MSE). (**b**) Reconstruction time.

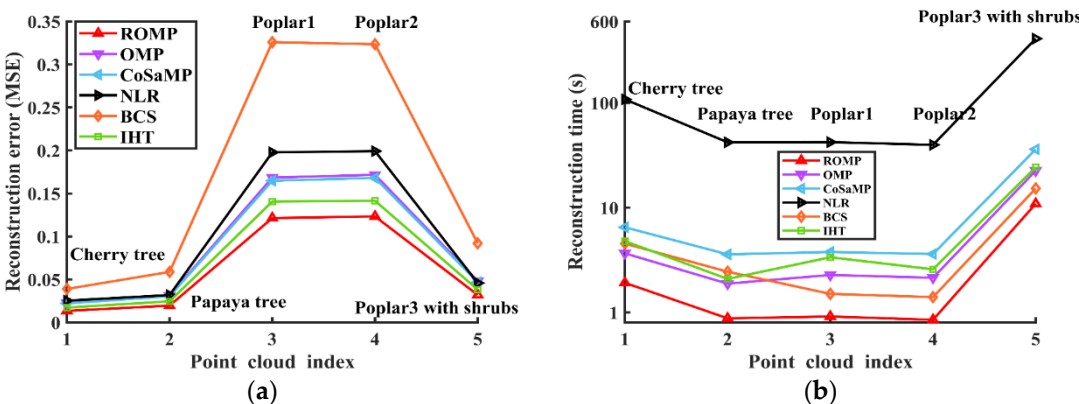

**Figure 10.** Comparison of reconstruction algorithms at compression ratio: 40%. (**a**) Reconstruction error (MSE). (**b**) Reconstruction time. Note that OMP denotes Orthogonal Matching Pursuit, CoSaMP denotes Compressive Sampling Matching Pursuit, NLR denotes Nonlocal Low-Rank Regularization, BCS denotes Bayesian Compressed Sensing, and IHT denotes Iterative Hard Thresholding.

In summary, the choice of Haar wavelet, partial Fourier observation matrix, and ROMP reconstruction algorithm in the proposed scheme are advantageous in handling broad-leaved tree point clouds, both in terms of reconstruction error and computation time.

### 3.2. Experiments on Single-Tree Point Clouds

In this set of experiments, point clouds (a)–(e) are used to verify the performance of the proposed scheme on point clouds from a single tree.

The main parameters of voxel filtering and statistical filtering in the preprocessing step are tuned separately for each point cloud in order to obtain the best performance. The results of filtering are shown in Tables 2 and 3.

**Table 2.** The results of statistical filtering.

| Point Clouds | Date Size before Filtering | Date Size after Filtering |
|---|---|---|
| *Cherry tree* | 293,631 | 286,937 |
| *Papaya tree* | 114,322 | 111,690 |
| *Poplar 1* | 146,635 | 146,015 |
| *Poplar 2* | 137,100 | 136,544 |
| *Poplar 3 with shrubs* | 1,349,406 | 1,344,029 |

**Table 3.** The results of voxel filtering.

| Point Clouds | Date Size before Filtering | Date Size after Filtering |
|---|---|---|
| *Cherry tree* | 286,937 | 274,502 |
| *Papaya tree* | 111,690 | 110,729 |
| *Poplar 1* | 146,015 | 117,268 |
| *Poplar 2* | 136,544 | 110,335 |
| *Poplar 3 with shrubs* | 1,344,029 | 1,053,598 |

The experimental results with different compression ratios are summarized in Table 4.

**Table 4.** Experiments on single-tree point clouds.

| Point Clouds | Compression Ratio | Reconstruction Time (s) | Reconstruction Error (MSE) |
|---|---|---|---|
| *Cherry tree* (274,502 × 3) | 20% | 2.9435 | 0.0079 |
|  | 40% | 1.9146 | 0.0135 |
|  | 60% | 1.6454 | 0.0175 |
|  | 80% | 1.1322 | 0.0329 |
| *Papaya tree* (110,729 × 3) | 20% | 1.5268 | 0.0123 |
|  | 40% | 0.8760 | 0.0196 |
|  | 60% | 0.6759 | 0.0279 |
|  | 80% | 0.4944 | 0.0521 |
| *Poplar 1* (117,268 × 3) | 20% | 1.4454 | 0.0830 |
|  | 40% | 0.9120 | 0.1214 |
|  | 60% | 0.6284 | 0.1896 |
|  | 80% | 0.4874 | 0.2795 |
| *Poplar 2* (110,335 × 3) | 20% | 1.2941 | 0.0779 |
|  | 40% | 0.8483 | 0.1233 |
|  | 60% | 0.5896 | 0.1940 |
|  | 80% | 0.4633 | 0.2890 |
| *Poplar 3 with shrubs* (1,053,598 × 3) | 20% | 15.1236 | 0.0184 |
|  | 40% | 10.8942 | 0.0320 |
|  | 60% | 8.7479 | 0.0482 |
|  | 80% | 7.8575 | 0.0995 |

The morphological features of the reconstructed point clouds are illustrated in Figures 11–15.

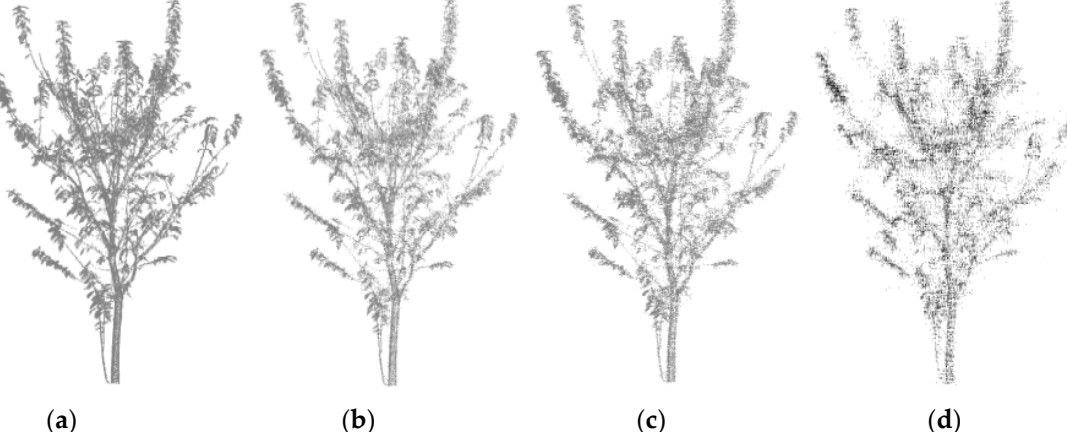

(**a**)　　　　　(**b**)　　　　　(**c**)　　　　　(**d**)

**Figure 11.** Compressed sensing (CS) recovered *Cherry tree* point cloud with different compression ratios. (**a**) Compression ratio: 20%, (**b**) compression ratio: 40%, (**c**) compression ratio: 60%, (**d**) compression ratio: 80%.

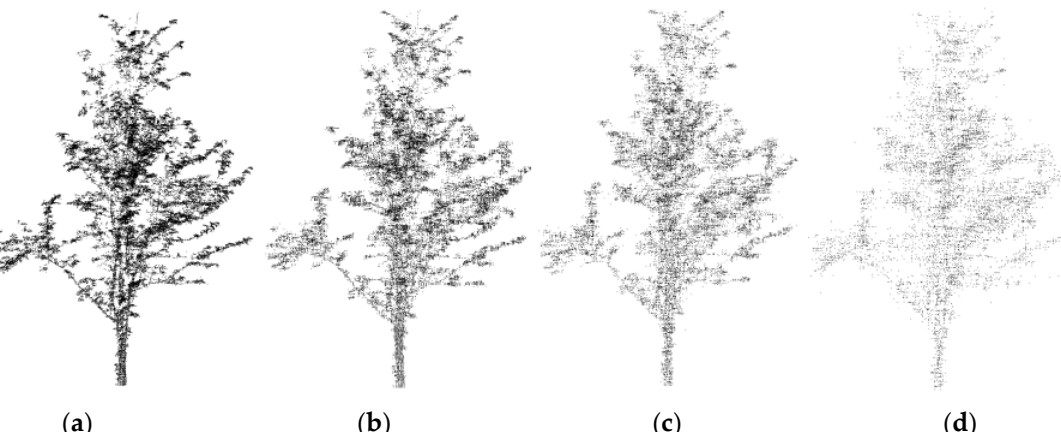

(**a**)　　　　　(**b**)　　　　　(**c**)　　　　　(**d**)

**Figure 12.** CS recovered *Papaya tree* point cloud with different compression ratios. (**a**) Compression ratio: 20%, (**b**) compression ratio: 40%, (**c**) compression ratio: 60%, (**d**) compression ratio: 80%.

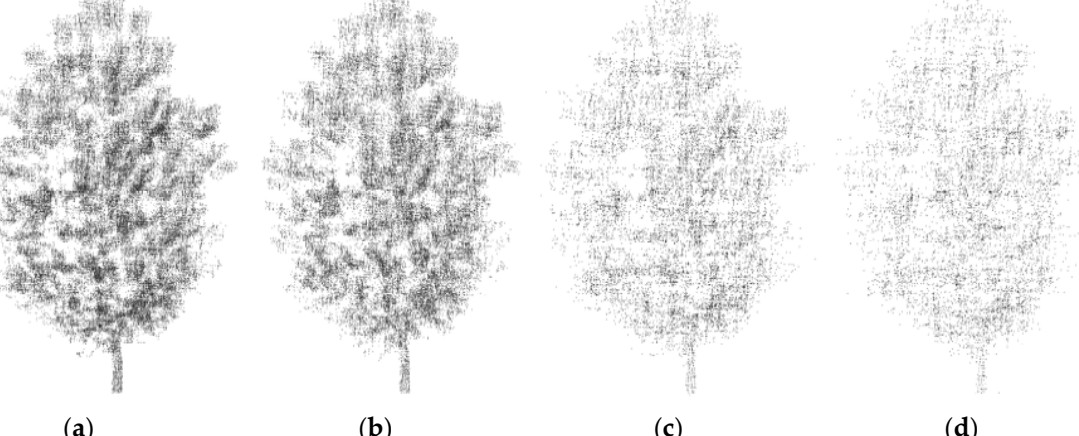

(**a**)　　　　　(**b**)　　　　　(**c**)　　　　　(**d**)

**Figure 13.** CS recovered *Poplar 1* point cloud with different compression ratios. (**a**) Compression ratio: 20%, (**b**) compression ratio: 40%, (**c**) compression ratio: 60%, (**d**) compression ratio: 80%.

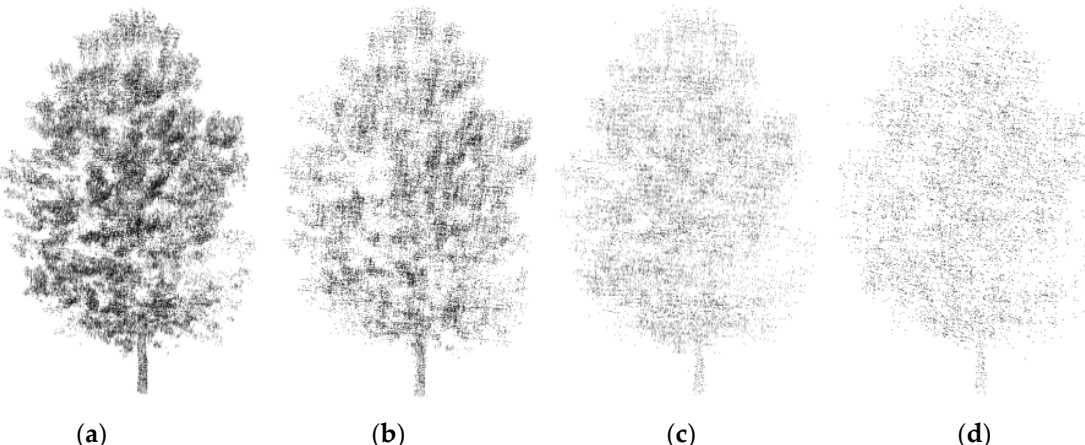

**(a)**　　　　　　**(b)**　　　　　　**(c)**　　　　　　**(d)**

**Figure 14.** CS recovered *Poplar 2* point cloud with different compression ratios. (**a**) Compression ratio: 20%, (**b**) compression ratio: 40%, (**c**) compression ratio: 60%, (**d**) compression ratio: 80%.

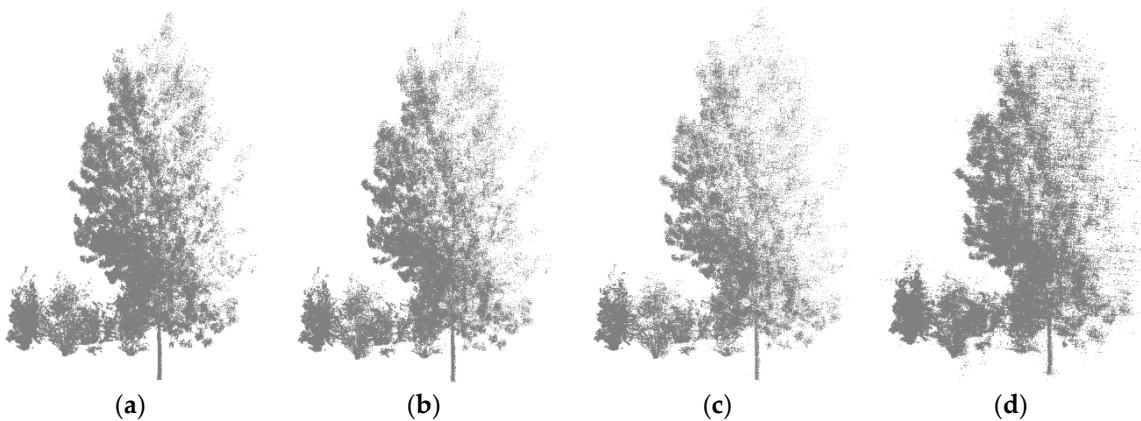

**(a)**　　　　　　**(b)**　　　　　　**(c)**　　　　　　**(d)**

**Figure 15.** CS recovered *Poplar 3 with shrubs* point cloud with different compression ratios. (**a**) Compression ratio: 20%, (**b**) compression ratio: 40%, (**c**) compression ratio: 60%, (**d**) compression ratio: 80%.

### 3.3. Experiments on Morphological Attributes

To further understand if the proposed scheme sufficiently preserves the original characteristics of a single tree, we quantify the morphological attributes of the reconstructed trees at different compression ratios. The morphological attributes considered include crown size (east-west, south-north, height) [4], diameter at breast height (DBH, diameter of the trunk at 1.30 m above the ground-surface) [2–4], ground diameter (diameter of the trunk at 0.20 m above the ground-surface, only for *Papaya tree*), tree height [2,4,5], which are important parameters for tree growth [1–5]. These morphological attributes are estimated with the software CloudCompare [42], a professional tool for processing point cloud. The results are summarized in Table 5.

For the *Papaya tree* point cloud, we estimate the ground diameter rather than DBH due to its special branch structures (Table 6). For simplicity, we assume that the cross section of the trunk is circular and apply least-squares circle fitting [3] to estimate DBH and the ground diameter. The relative error statistics compared to measurements from uncompressed point clouds are also given in Table 7.

**Table 5.** Experiments on morphological attributes.

| Point Clouds | Compression Ratio | Crown Size (East-West, South-North, Height) (m) | Diameter at Breast Height (DBH, m) | Tree Height (m) |
|---|---|---|---|---|
| *Cherry tree* | 0% | 3.0520, 3.0263, 4.1213 | 0.0704 | 5.0193 |
| | 20% | 3.0624, 3.0244, 4.1265 | 0.0700 | 5.0276 |
| | 40% | 3.0591, 3.0306, 4.1309 | 0.0691 | 5.0202 |
| | 60% | 3.0475, 3.0384, 4.1357 | 0.0700 | 5.0223 |
| | 80% | 3.0423, 3.0741, 4.1441 | 0.0701 | 5.0363 |
| *Poplar 1* | 0% | 10.4139, 10.3026, 15.9860 | 0.3665 | 19.0771 |
| | 20% | 10.4625, 10.2566, 15.9743 | 0.3700 | 19.1161 |
| | 40% | 10.4422, 10.2605, 15.9552 | 0.3735 | 19.1176 |
| | 60% | 10.4044, 10.3855, 15.9098 | 0.3741 | 19.0294 |
| | 80% | 10.4829, 10.4011, 15.9433 | 0.3773 | 19.1235 |
| *Poplar 2* | 0% | 10.4491, 10.0871, 15.9337 | 0.3620 | 19.1354 |
| | 20% | 10.5182, 10.0796, 15.9818 | 0.3539 | 19.1475 |
| | 40% | 10.4058, 10.0800, 15.7993 | 0.3658 | 19.0596 |
| | 60% | 10.4786, 10.2239, 15.9947 | 0.3515 | 18.9318 |
| | 80% | 10.5838, 10.2361, 16.1414 | 0.3717 | 19.1148 |
| *Poplar 3 with shrubs* | 0% | 10.1693, 10.4654, 15.5625 | 0.3598 | 18.5915 |
| | 20% | 10.1940, 10.4629, 15.5920 | 0.3628 | 18.5740 |
| | 40% | 10.1370, 10.4903, 15.6360 | 0.3619 | 18.6114 |
| | 60% | 10.1766, 10.5377, 15.6412 | 0.3494 | 18.6125 |
| | 80% | 10.2543, 10.6840, 15.7492 | 0.3555 | 18.7761 |

**Table 6.** Experiment on morphological attributes of *Papaya tree.*

| Point Cloud | Compression Ratio | Crown Size (East-West, South-North, Height) (m) | Ground Diameter (m) | Tree Height (m) |
|---|---|---|---|---|
| *Papaya tree* | 0% | 2.4256, 3.1193, 4.6847 | 0.0442 | 5.1134 |
| | 20% | 2.4589, 3.1246, 4.7133 | 0.0438 | 5.1237 |
| | 40% | 2.4390, 3.1283, 4.7088 | 0.0455 | 5.1089 |
| | 60% | 2.4178, 3.1339, 4.7248 | 0.0449 | 5.1133 |
| | 80% | 2.4749, 3.1923, 4.6561 | 0.0457 | 5.1311 |

**Table 7.** The relative error statistics of morphological attributes.

| Point Clouds | Compression Ratio | Crown Size (East-West, South-North, Height) | DBH & Ground Diameter | Tree Height |
|---|---|---|---|---|
| *Cherry tree* | 20% | 0.3407%, 0.0628%, 0.1262% | 0.5682% | 0.1657% |
| | 40% | 0.2326%, 0.1421%, 0.2329% | 1.8466% | 0.0179% |
| | 60% | 0.1474%, 0.3998%, 0.3494% | 0.5682% | 0.0598% |
| | 80% | 0.3178%, 1.5795%, 0.5532% | 0.4261% | 0.3387% |
| *Poplar 1* | 20% | 0.4667%, 0.4465%, 0.0732% | 0.9550% | 0.2044% |
| | 40% | 0.2718%, 0.4086%, 0.1927% | 1.9099% | 0.2123% |
| | 60% | 0.0912%, 0.8047%, 0.4767% | 2.0737% | 0.2500% |
| | 80% | 0.6626%, 0.9560%, 0.2671% | 2.9468% | 0.2432% |
| *Poplar 2* | 20% | 0.6613%, 0.0743%, 0.3019% | 2.2376% | 0.0632% |
| | 40% | 0.4144%, 0.0704%, 0.8435% | 1.0497% | 0.3961% |
| | 60% | 0.2823%, 1.3562%, 0.3828% | 2.9006% | 1.0640% |
| | 80% | 1.2891%, 1.4771%, 1.3035% | 2.6796% | 0.1077% |
| *Poplar 3 with shrubs* | 20% | 0.2429%, 0.0239%, 0.2793% | 0.8338% | 0.0941% |
| | 40% | 0.3176%, 0.2379%, 0.6959% | 0.5837% | 0.1070% |
| | 60% | 0.0717%, 0.6908%, 0.7451% | 2.8905% | 0.1130% |
| | 80% | 0.8358%, 2.0888%, 1.7675% | 1.1951% | 0.9929% |
| *Papaya tree* | 20% | 1.3729%, 0.1699%, 0.6105% | 0.9050% | 0.2014% |
| | 40% | 0.5524%, 0.2885%, 0.5144% | 2.9412% | 0.0880% |
| | 60% | 0.3216%, 0.4681%, 0.8560% | 1.5837% | 0.0001% |
| | 80% | 2.0325%, 2.3403%, 0.6105% | 3.3937% | 0.3461% |

### 3.4. Extension to Plot-Level

To evaluate the performance of the proposed scheme on large scenes, we prepare 3 plot point clouds: *Sapindus plot*, *Poplar plot*, and *Rubber tree plot*. The detailed experimental results are given in Tables 8–10, and graphical displays are shown in Figures 16–18. We can see that similar to the cases of a single tree, the proposed scheme can achieve low reconstruction errors within reasonable reconstruction time.

**Table 8.** The results of statistical filtering.

| Point Clouds | Date Size before Filtering | Date Size after Filtering |
|---|---|---|
| *Sapindus plot* | 9,694,521 | 9,158,523 |
| *Poplar plot* | 12,824,963 | 11,535,794 |
| *Rubber tree plot* | 4,592,723 | 4,193,915 |

**Table 9.** The results of voxel filtering.

| Point Clouds | Date Size before Filtering | Date Size after Filtering |
|---|---|---|
| *Sapindus plot* | 9,158,523 | 7,836,739 |
| *Poplar plot* | 11,535,794 | 9,127,145 |
| *Rubber tree plot* | 4,193,915 | 2,852,350 |

**Table 10.** Experiments on plot point clouds.

| Point Clouds | Compression Ratio | Reconstruction Time (s) | Reconstruction Error (MSE) |
|---|---|---|---|
| *Sapindus plot* (7,836,739 × 3) | 20% | 295.8003 | 0.0063 |
| | 40% | 277.4267 | 0.0091 |
| | 60% | 268.4071 | 0.0129 |
| | 80% | 253.6288 | 0.0211 |
| *Poplar plot* (9,127,145 × 3) | 20% | 366.7588 | 0.0961 |
| | 40% | 355.7145 | 0.1226 |
| | 60% | 346.3504 | 0.1687 |
| | 80% | 336.7655 | 0.2245 |
| *Rubber tree plot* (2,852,350 × 3) | 20% | 48.2435 | 0.0719 |
| | 40% | 44.5872 | 0.0903 |
| | 60% | 41.6949 | 0.1247 |
| | 80% | 39.4404 | 0.1405 |

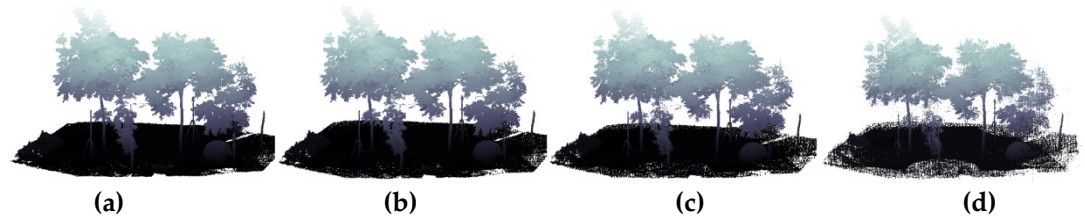

|     (a)     |     (b)     |     (c)     |     (d)     |

**Figure 16.** CS recovered *Sapindus plot* point cloud with different compression ratios. (**a**) Compression ratio: 20%, (**b**) compression ratio: 40%, (**c**) compression ratio: 60%, (**d**) compression ratio: 80%.

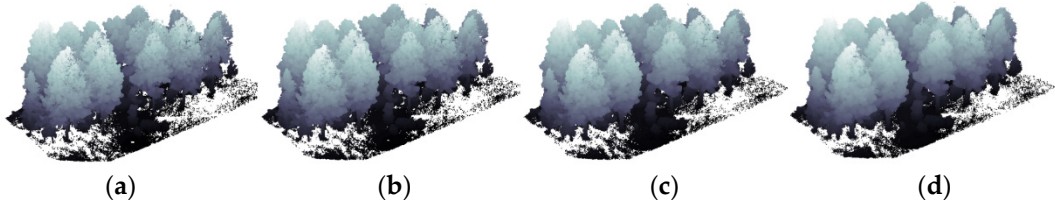

**Figure 17.** CS recovered *Poplar plot* point cloud with different compression ratios. (**a**) Compression ratio: 20%, (**b**) compression ratio: 40%, (**c**) compression ratio: 60%, (**d**) compression ratio: 80%.

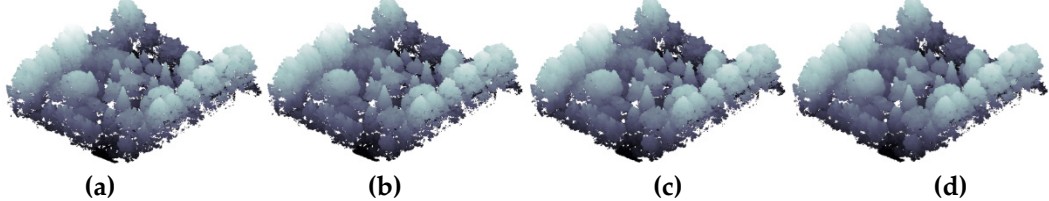

**Figure 18.** CS recovered *Rubber tree plot* point cloud with different compression ratios. (**a**) Compression ratio: 20%, (**b**) compression ratio: 40%, (**c**) compression ratio: 60%, (**d**) compression ratio: 80%.

## 4. Discussion

### 4.1. Effects of Voxel Filtering and Statistical Filering

#### 4.1.1. Statistical Filtering

When collecting point clouds, outliers can be generated due to measurement noise. Such noises tend to be sparsely distributed. It is important to remove outliers since they can reduce the accuracy of reconstruction. In statistical filtering for outlier removal, one needs to select a search radius and a threshold. There exists a trade-off between outlier removal and retention of normal points. With a larger search radius, more outliers can be removed, and with a smaller threshold, more normal points can be removed as outliers. In this study, the appropriate search radius and threshold are 10–50 points and 0.5–5, respectively. The statistical filtering is realized by PCL (Point Cloud Library) [43,44]

As an example, in the *Cherry tree* point cloud, statistical filtering eliminates 12,435 abnormal points (Figure 19).

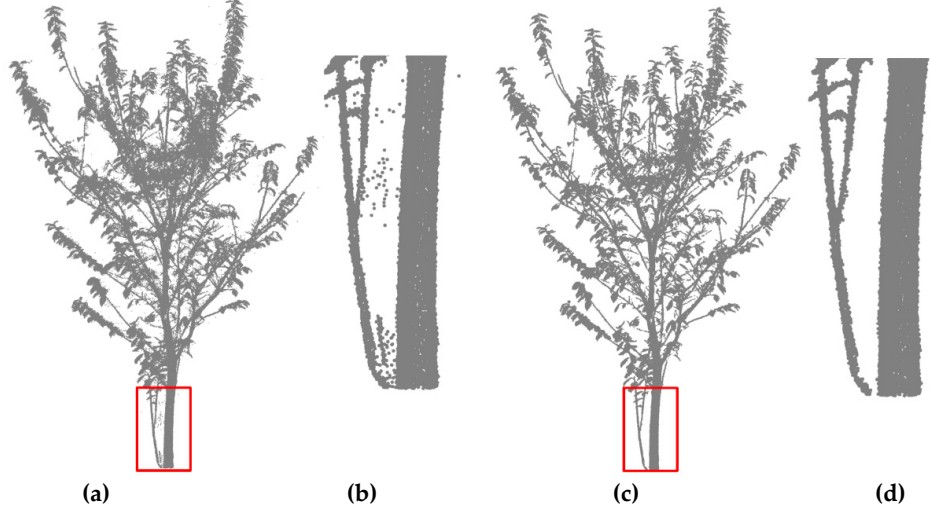

**Figure 19.** Statistical filtering removes outliers. (**a**) Original *Cherry tree* point cloud. (**b**) Details of the underneath trunk before statistical filtering. (**c**) Statistical filtered *Cherry tree* point cloud. (**d**) Details of the underneath trunk after statistical filtering.

4.1.2. Voxel Filtering

Voxel filtering is a method to simplify point clouds. It can reduce the size of point clouds while maintaining the original geometric structure of point clouds. It is mostly used in the preprocessing of dense point clouds and have been applied in surface reconstruction, shape recognition, etc. A point cloud can be down-sampled depending on the size of voxels. With larger voxels, the filtered point cloud is thinner (Figure 20). A small voxel size is suitable for studying the local morphological attributes of a tree, while a large voxel size makes it easier to observe the global growth of a plot. Hence, the choice of voxel size is application-specific. In this study, voxel filtering is applied to simplify point clouds in order to accelerate the speed of subsequent operations. A voxel size of 0.005 m–0.02 m is reasonable for most broad-leaved trees. Voxel filtering is also realized by PCL in our implementation.

Take the point cloud *Poplar 3 with shrubs* as an example. Voxel filtering removes 290,431 points, equivalent to reduce the data size by 22%. This leads to a 0.5 s–1 s reduction in computation time for a small point cloud like *Cherry tree*, and 3 s–5 s for a large point cloud such as *Poplar 3 with shrubs*, without any negative effect on the morphological characteristics of trees.

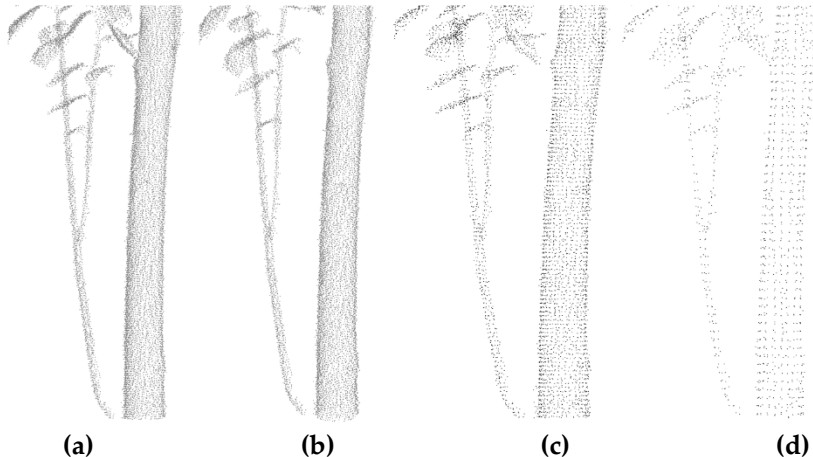

**(a)**                 **(b)**                 **(c)**                 **(d)**

**Figure 20.** The voxel filtering only dilutes the point clouds and does no damage to morphological attributes. (**a**) *Cherry tree* point cloud after statistical filtering. (**b**) Voxel filtering on *Cherry tree* point cloud with a voxel size of 0.005 m. (**c**) Voxel filtering on *Cherry tree* point cloud with a voxel size of 0.01 m. (**d**) Voxel filtering on *Cherry tree* point cloud with a voxel size of 0.02 m.

*4.2. Justification for Choosing Sparse Structure and Selection of Sparse Basis*

In the past 10 years, the requirement of compressed sensing for data has changed from general sparsity to specific structures, such as low rank [45,46], group sparsity [47,48] and so on. Compressed sensing based on low rank and group sparsity has been widely employed in image processing due to their high compression ratio and reconstruction accuracy. Nevertheless, point clouds have different characteristics compared to general images. For example, all points have distinctive XYZ coordinates. Point clouds tend to be cluttered and disordered, making low rank and group sparsity structure unable to achieve satisfactory performances. Moreover, we find that most data are distributed in a roughly step-wise pattern, like the shape of Haar wavelet. This phenomenon prompts us to choose Haar wavelet as sparse basis.

We also design an experiment to confirm that the Haar wavelet is indeed superior to other classical wavelets, such as db2, sym2, coif1, bior1.1, rbio1.1, fk4. The experiment illustrates that Haar wavelet has the highest reconstruction accuracy among the bases tested on five single-tree point clouds with comparable reconstruction time. This confirms the observation of step-wise patterns in the data, and Haar wavelet as a suitable wavelet basis.

### 4.3. Selection of Observation Matrix

Random Gaussian matrix [49,50] was proved to be the most universal observation matrix in compressed sensing because it is uncorrelated with most orthogonal bases or dictionaries. Later, the partial Fourier matrix [51–53] was proposed to replace the random Gaussian matrix. Over time, more and more special structured matrices that satisfied RIP were proposed for observation matrices, including sparse random matrix [49,53], partial Hadamard matrix [53], Bernoulli matrix [49,54], Toeplitz matrix [55–58], and Circulant matrix [57,58].

Random Gaussian matrix is uncorrelated with most orthogonal bases of dictionaries and can meet the requirement of accurate reconstruction when $M \geq cK \log(N/K)$, where M, N denote the dimensions of the observation matrix, K is sparse level of the data and c is a constant. Partial Fourier matrix also satisfies RIP, but the sparse level K must conform to $K \leq cM(\log(N))^6$. Sparse random matrix and Bernoulli matrix stand out because they are easy to implement and store. Partial Hadamard matrix requires a smaller number of measurements due to its uncorrelation and partial orthogonality. However, its dimension must meet $N = 2^k$, $k = 1, 2, 3,\ldots$, which greatly limits its application. Toeplitz matrix and Circulant matrix are attractive as they are easy to implement in hardware.

From Figure 9, we find that the reconstruction errors of partial Fourier matrix are much lower than that of other observation matrices. For small point clouds such as *Cherry tree* and *Papaya tree*, the reconstruction errors using partial Fourier matrix are 10%–20% smaller than those using other observation matrices, and for a large point cloud like *Poplar 3 with shrubs*, the error is 16%–21% smaller than others. However, the smaller reconstruction error comes at the expense of longer reconstruction time. For *Cherry tree*, *Papaya tree*, *Poplar 1*, *Poplar 2*, the reconstruction time using partial Fourier matrix is 1.5 times that of other observation matrices. For *Poplar 3 with shrubs*, the reconstruction time is 2 times more. The desired trade-off between accuracy and computation time is application-dependent. In forest inventories, the time cost is acceptable since accuracy is important in preserving tree morphological structures. However, partial Fourier matrix may not be suitable for extremely large point clouds due to its higher complexity. In this case, other observation matrices which also have a good performance on broad-leaved tree point clouds like Circulant matrix can be adopted.

### 4.4. Selection of Reconstruction Algorithm

Reconstruction in compressed sensing is first formulated as an optimization problem. Many reconstruction algorithms have been proposed in literature, including MP (Matching Pursuit) [59], OMP (Orthogonal Matching Pursuit) [60,61], CoSaMP (Compressive Sampling Matching Pursuit) [62], ROMP (Regularized Orthogonal Matching Pursuit) [41], HTP (Hard Thresholding Pursuit) [63], IHT (Iterative Hard Thresholding) [64], etc. Several algorithms have also been proposed to solve the reconstruction problem as an estimation problem, e.g., BCS (Bayesian Compressed Sensing) [65], AMP (Approximate Message Passing) [66], etc. Later efforts focused on solving problems with specific structures, e.g., NLR (Nonlocal Low-Rank Regularization) [67] and SGSR (Structural Group Sparse Representation) [48]. Nowadays, combining compressed sensing with the popular deep learning, the deep-CS [68–77] becomes the focus of current research. Owing to the complex structures of trees, it is challenging to select an algorithm that satisfies both criterions of low computation time and high accuracy. Moreover, most algorithms, such as NLR, have many parameters that are difficult to tune while others, such as deep-CS, rely too much upon abundant training data. Based on these observations, we select the classical ROMP algorithm, which can achieve a high accuracy in a short computation time with few tunable parameters.

In the experiment, the ROMP algorithm has a better performance in both reconstruction errors and time. The ROMP algorithm outperforms other algorithms by 22%–63% in terms of MSE, and 13%–97% in terms of reconstruction time. Therefore, we conclude that the ROMP algorithm is the best algorithm in reconstructing broad-leaved tree point clouds among all algorithms experimented.

*4.5. Impacts on Morphological Attributes*

A single tree has many significant morphological attributes, such as crown size, DBH, tree height, etc. Good compression and reconstruction algorithms should have negligible impacts on these attributes. Figures 21–23 illustrate crown size, DBH and tree height under different compression ratios. Tables 5 and 6 compare these attributes estimated from reconstructed and original point clouds.

From Tables 5 and 6, we observe that the crown size, DBH and tree height fluctuate around the actual value. The relative error statistics are summarized in Table 7. The relative errors range from 0.0010%–3.3937%, which is acceptable in our application. Next, we provide an in-depth analysis on the errors.

Firstly, in forest inventories, it is common to use one terrestrial laser scanner to scan a wide area. Missing data is common in this case, especially at the back of the trees (Figure 24). This makes it difficult to compute morphological attributes such as DBH, because the cross sections of tree trunks are incomplete (Figure 25). Missing data may influence fitting and lead to the miscalculation of DBH. As evident from the experimental results, our scheme can handle this situation and determine DBH within acceptable errors.

Secondly, to improve the estimation accuracy of morphological attributes, we perform an extra statistical filtering on the reconstructed data to eliminate outliers. However, due to missing data during acquisition, some parts such as branches appear as if they are torn apart from the trunk. When the compression ratio is very high, these points are sparser and as a result, statistical filtering will eliminate them as outliers. This may cause a slight reduction in estimated morphological attributes, e.g., the crown size (south-north) of *Cherry tree* when the compression ratio is 80%.

Finally, it is expected that the larger reconstruction errors will reduce estimation accuracy. As the compression ratio increases, more point clouds deviate from their original positions. If a point far from its original position happens to be retained by statistical filtering, there will be a slight increase of morphological attributes as indicated by the crown size and tree height of *Poplar 3 with shrubs* when compression ratio is 80%. On the other hand, when reconstruction causes overlaps of points and thus some local points may be missing, morphological attributes may be underestimated as indicated by the sudden decline of tree height of *Poplar 2* when compression ratio is 60%.

In conclusion, compression and reconstruction will affect morphological attributes to a certain degree. As compression ratio increases, the impact tends to be bigger. However, the overall error ranges from 0.0010%–3.3937%, which will not have a considerable influence on practical applications.

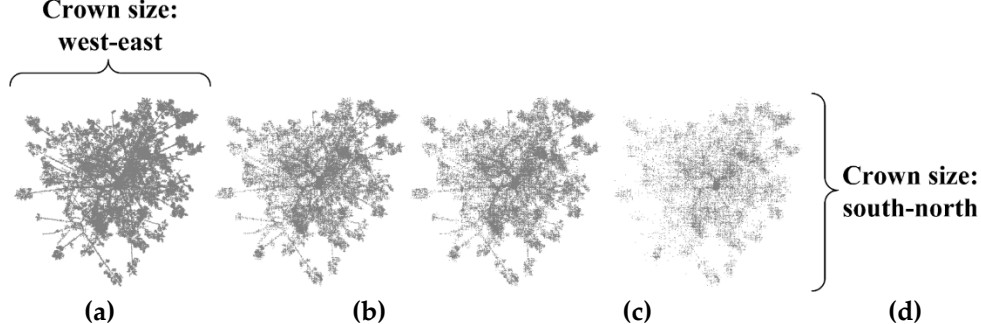

**Figure 21.** The crown size (west-east and south-north) of *Cherry tree* with different compression ratios. (**a**) Compression ratio: 20%, (**b**) compression ratio: 40%, (**c**) compression ratio: 60%, (**d**) compression ratio: 80%.

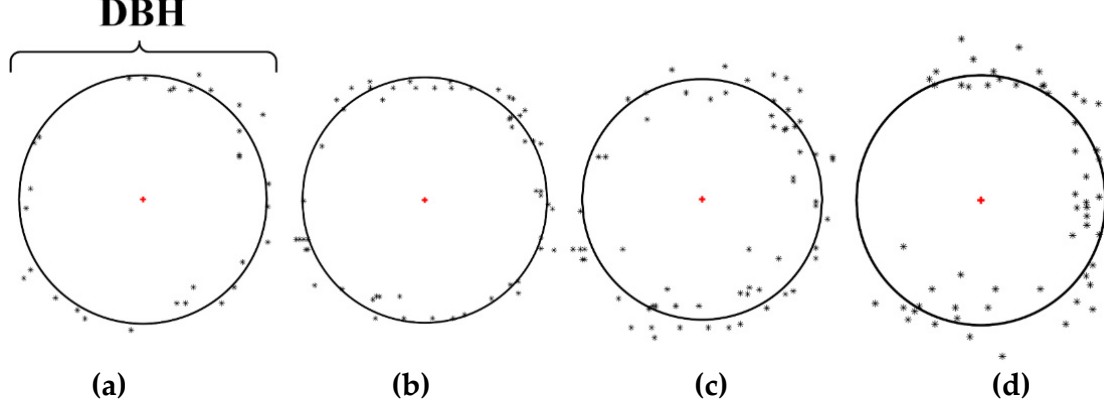

**Figure 22.** The DBH of *Cherry tree* with different compression ratios. (**a**) Compression ratio: 20%, (**b**) compression ratio: 40%, (**c**) compression ratio: 60%, (**d**) compression ratio: 80%.

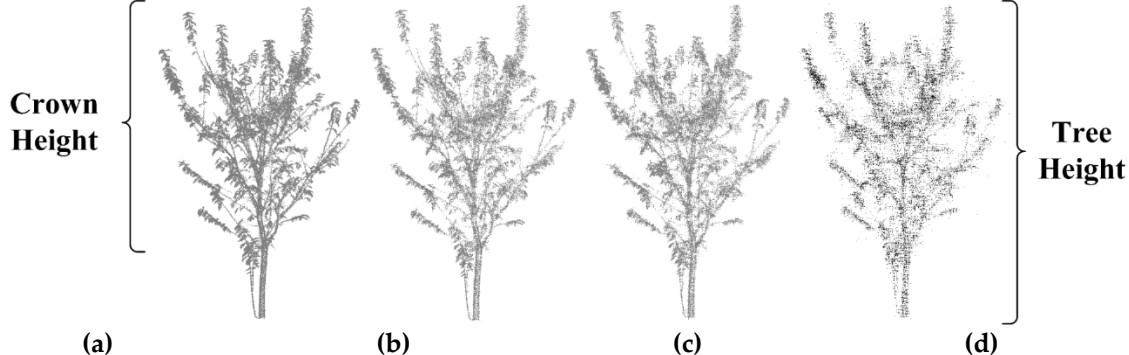

**Figure 23.** The crown height and tree height of *Cherry tree* with different compression ratios. (**a**) Compression ratio: 20%, (**b**) compression ratio: 40%, (**c**) compression ratio: 60%, (**d**) compression ratio: 80%.

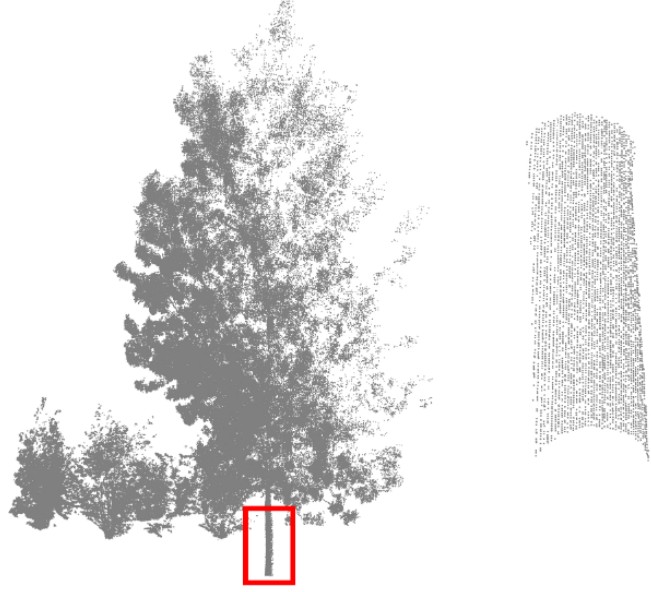

**Figure 24.** Incomplete tree trunk of *Poplar 3 with shrubs* caused by data collection method.

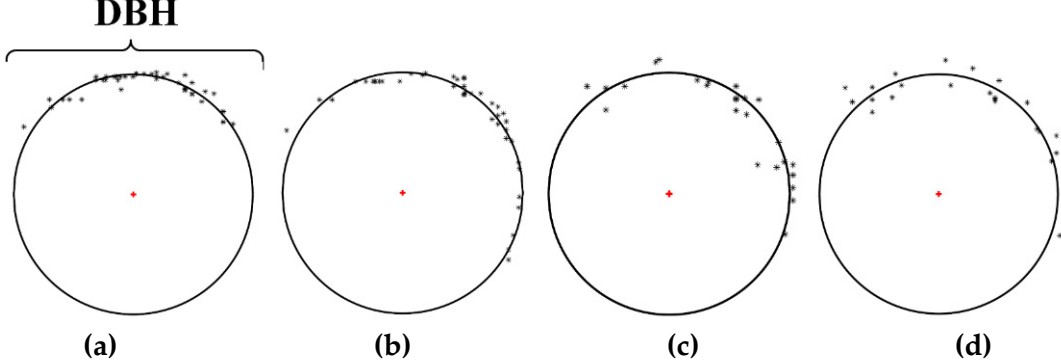

**Figure 25.** The DBH of *Poplar 3 with shrubs* with different compression ratios. (**a**) Compression ratio: 20%, (**b**) compression ratio: 40%, (**c**) compression ratio: 60%, (**d**) compression ratio: 80%.

### 4.6. Generalization to Plot-Level

Experimental results show that the proposed scheme has an acceptable performance on plot point clouds. A few comments are in order. First, the choice of sparse basis depends on the characteristics of the point cloud very much. For both single-tree and plot-level point clouds, we find strong correlations with the Harr wavelet basis. This explains the excellent performance of the Harr wavelet. Second, as evident from the results, the time complexity of compression using partial Fourier matrix is higher than other matrices. Though it is acceptable for small point clouds, random Gaussian matrix or Circulant matrix are better candidates for plot-level point clouds since they achieve a good trade-off between computation time and accuracy. Lastly, it appears that dense or high-quality point clouds have a better reconstruction performance. This may be attributed to two reasons, namely, (1) high redundancy, and (2) sufficient resolutions to learn sparse structures properly.

In conclusion, the proposed scheme can indeed handle different plot structures with acceptable MSE at 0.0063–0.2245, demonstrating its generalization to plot-level.

### 4.7. Future Work

As the current scheme fails to reconstruct RGB information of broad-leaved tree point clouds, one interesting extension to take RGB information into consideration. Additionally, in the experiments, we find when compression ratio exceeds a critical value, the reconstructed broad-leaved tree point clouds tend to be sparse due to the overlapping points. For further improvement, we plan to investigate smoothing or filled function. Another venue of future work is to generalize the proposed scheme to other categories of trees, such as conifers, and larger application scenes with various forest structures, such as mingled forests. Special sparse transform matrix, observation matrix, and reconstruction algorithm will be designed to optimize the MSE and reconstruction time based on the characteristics of the target data.

## 5. Conclusions

In this paper, a compressed sensing scheme has been proposed to compress and reconstruct broad-leaved tree point clouds. Unlike conventional point cloud compression algorithms that perform compression after full sampling, the proposed scheme completes compression in the process of sampling, which greatly reduces computation time and storage space. Moreover, instead of considering the topological relationship among points, spatial coordinate information was compressed directly based on the characteristics of broad-leaved tree point clouds. We identified the most suitable sparse transformation matrix, observation matrix, and reconstruction algorithm by analyzing and empirically comparing different approaches. Experimental results demonstrated that the proposed scheme achieves superior performances on both single-tree and plot-level point clouds. The significant reduction in

data volume makes it possible for real-time transmission and storage of broad-leaved tree point clouds in forest inventories.

**Supplementary Materials:** The following are available online at http://www.mdpi.com/1999-4907/11/3/257/s1, PDF S1: A Brief Introduction to Compressed Sensing; Excel S1: Experimental result.

**Author Contributions:** R.X. devised the programs and drafted the initial manuscript. T.Y. helped with data collection, data analysis, and figures and tables. L.C. contributed to fund acquisition and writing embellishment. Y.L. designed the project and revised the manuscript. All authors have read and agreed to the published version of the manuscript.

**Acknowledgments:** This research was funded by the National Key R&D Program of China (grant number 2017YFD0600904) and the Priority Academic Program Development of Jiangsu Higher Education Institutions (PAPD). Appreciate Chunhua Hu from Nanjing Forestry University for providing data, Henny Lee from Nanjing Forestry University and Rong Zheng from McMaster University for providing revision advice.

**Conflicts of Interest:** The authors declare no conflict of interest.

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
