# Peer review of "Compression and Recovery of 3D Broad-Leaved Tree Point Clouds Based on Compressed Sensing"

_forests, doi:10.3390/f11030257_

Round 1

Reviewer 1 Report

i am concerned about the change in tune of the article. First, the authors do not provide a point-by-point response to the reviewers, so it is impossible to see what it was suggested and how it was resolved. This is not acceptable. Second, I have a high degree of concern about the change on the conclusion where the authors go from indicating that their method has problems to a total different conclusion. It seems to me that that the change in the conclusion has been done only to satisfy the fact the the paper will be published which is not correct. 

Reviewer 2 Report

I only had a few minor comments on the first version of the manuscript. The authors have addressed them all and they have significantly improved their manuscript. It is robust and its the findings will be useful for the scientific community involved with the use of such data.

Reviewer 3 Report

Line 29: You said that the TLS is cost efficient. In what way do you refer that the costs are low? costs regarding the time, the first investment for the equipment, or the human resource? There are similar studies regarding the cost efficiency and they do not necessarily conclude that TLS is very cost efficient, at least when studying a large area.

Line 188: On the producer`s page it is advertised that this sensor generates a point cloud of up to 695,000 points per second with a range of up to 100 m and a typical accuracy of ±2 cm, how did your obtained 700.000?

Line 188: For validating the ±2 cm accuracy do you refer to the absolute accuracy, or just expressed as RMSE?

Line 213: You mention using MATLAB, but is that software robust enough to process large datasets? or you just made the analysis on a small data set? For example, is it capable to process a data set for a large forest scan?

Line 529: Why after filtering, the rubber tree plot reduced its size that much, comparing it to the other experimental plots?

Author Response

This manuscript is a resubmission of an earlier submission. The following is a list of the peer review reports and author responses from that submission.

Round 1

Reviewer 1 Report

Laser Scanners, either terrestrial or aerial, represent a great advancement in remote sensing with a wide range of application possibilities, from generating forest profiles to creating relief digital representations among others. This manuscript is a technical one and presents a promising methodological approach for dealing with the issue of data transfer and processing, which often prevent the application of this advanced technology in large areas or complex forms. The manuscript reads well and the few minor linguistic mistakes I am sure they will be corrected in the revised version of the manuscript.

The introduction provides the necessary background information and it has a clearly stated problem statement which is based on previous research. The methods employed are sound and adequately described in a manner that allows their transferabillity to other datasets. The results are presented with a good combination of figures, tables and graphs which allow the inspection by the reader of the differences in performance as well as the produced error when compression rate is increasing and processing time decreases.  The discussion is based on the results achieved with no speculation while the limitations of the proposed methodology are also mentioned.

Reviewer 2 Report

The authors present an compression and reconstruction algorithm for TLS of single trees. The authors, in the introduction provide a justification of their work based on the current literature and provide a detail description of their technique. The manuscript has potential, but I am concerned about several elements:

Although the algorithm seems to work fine on single trees there is no indication of its success at the plot level. It will be important even to have a test at this level (plot) to see if it can be generalized. The paper lacks a significant review of the english. Some sections are difficult to follow and others some figures have spelling errors. There are many mega-sentences on the manuscript which make it very hard to read. There are significant assumptions that are not proven on the manuscript, for example on line 261 the authors indicate  that there is no impact on morphological features, but this is not proven on the paper? How elements like fractal geometry, crown size, DBH and tree height changes as function of the compression level? I am concerned about some of the comments from the authors regarding future work, it seems based on what they write that their algorithm is not 100% ready and there is a significant amount of work that needs to be done before it can be published. As such, there are two elements that must be considered: 1) Extent to the plot level, and 2) evaluate the impact on morphological features. One thing is that the algorithm can compress data and another is the impact of compression on elements associated to morphological features.
